# HBsAg Dampened STING Associated Activation of NK Cells in HBeAg-Negative CHB Patients

**DOI:** 10.3390/ijms22147643

**Published:** 2021-07-16

**Authors:** Bingqing Zheng, Yating Yu, Zhaoyi Pan, Yujie Feng, Huajun Zhao, Qiuju Han, Jian Zhang

**Affiliations:** Institute of Immunopharmaceutical Sciences, School of Pharmaceutical Sciences, Shandong University, Jinan 250012, China; zhengbq880223@126.com (B.Z.); marchyuu@163.com (Y.Y.); zhaoypan@163.com (Z.P.); yujfeng94@163.com (Y.F.); huajunzhao08@126.com (H.Z.); hanqiuju@sdu.edu.cn (Q.H.)

**Keywords:** NK cells, STING, STAT3, CHB

## Abstract

NK cells play crucial roles in defending against persistent HBV. However, NK cells present dysfunction in chronic hepatitis B virus (CHB) infection, and the associated mechanism is still not fully understood. Except for the regulatory receptors, NK cells could also be regulated by the surface and intracellular pattern recognition receptors (PRRs). In the present study, we found that the level of the adaptor of DNA sensor STING in NK cells was significantly decreased in HBeAg-negative CHB patients, and it was positively associated with the degranulation ability of NK cells. Compared to NK cells from healthy donors, NK cells from HBeAg-negative CHB patients displayed a lower responsiveness to cGAMP stimulation. Further investigation showed that HBsAg could inhibit the STING expression in NK cells and suppress the response of NK cells to cGAMP. Significantly, STAT3 was identified to be a transcription factor that directly regulated STING transcription by binding to the promoter. In addition, STAT3 positively regulated the STING associated IFN-α response of NK cells. These findings suggested that STING is an important adaptor in NK cell recognition and activation, while HBsAg disturbs NK cell function by the STAT3-STING axis, providing a new mechanism of NK disability in HBeAg-negative CHB infection.

## 1. Introduction

Hepatitis B virus (HBV) is DNA pathogenic virus that can cause chronic hepatitis, which is one of the major risk factors of liver fibrosis, cirrhosis and hepatocellular carcinoma (HCC). Despite the success of HBV vaccines, there are still 240 million chronic HBV (CHB) carriers in the world [1,2]. The current clinical treatment of CHB patients only achieves virus suppression and hepatitis B envelope antigen (HBeAg) sero-clearance, but rarely hepatitis B surface antigen (HBsAg) loss [3], so that high risk of CHB progressing to HCC still exists, and the underlying immunoregulatory mechanism of which has not been completely clarified [4,5].

The innate immune system is the first line for defense against virus invasion, which recognizes pathogen-associated molecular patterns (PAMP) by a series of pattern recognition receptors (PRRs) [6,7,8,9]. However, the adaptive immune responses could be detected weeks after HBV infection, which delays the optimal opportunity for diagnosis and treatment of HBV infection. Studies have indicated that DNA and RNA recognition signals of the innate immune system were injured during chronic HBV infection, especially the downregulation of toll-like receptors (TLRs) and the dysfunction of the type I IFN response, inducing persistent HBV [10,11,12,13]. HBV compositions, including HBV polymerase and HBV X (HBx) protein, could directly inhibit the type I IFN response by disrupting the phosphorylation of IFN regulatory factor 3 (IRF3) [13,14]. In addition, HBsAg and HBeAg could inhibit TLR2 and TLR4 signaling-associated immune response of dendritic cells and monocytes [15,16,17].

Natural killer (NK) cells are an essential component of the innate immune system, and they exhibit multiple functions to eliminate virus-infected cells, but the activities of NK cells are impaired in CHB patients. The levels of NKG2D and 2B4, activating receptors of NK cells, were found to be downregulated on NK cells from CHB patients, resulting in the impairment of NK cell functions, such as IFN-γ production and cytotoxicity [18]. It was shown that HBsAg inhibited the expression of MICA and MICB during CHB infection, dampening the activating receptor NKG2D-induced NK cell activation [19]. In contrast, NKG2A, one of the inhibitory receptors of NK cells, was upregulated on NK cells, accompanied with the increase of serum TGF-β and IL-10 levels in CHB patients [20]. However, the NK cell is not only regulated by the surface regulatory receptors, but it also can be activated by the PRR-associated signal pathways, such as TLRs, RIG-I-like receptors (RLRs) and cGAS DNA sensors [21,22]. Numerous studies have indicated that stimulation of TLRs, including TLR2, TLR3, TLR5, TLR7 and TLR9, could induce IFN-γ production and enhance cytotoxicity of NK cells [21]. Combined with the treatment of Nucleos(T)Ide Analogues, TLR7 agonist GS-9620 improved NK cell expansion and the anti-HBV effect, but reduced HBV-induced suppression of T-cell response in CHB patients [23]. In our previous study, we illustrated that HBV transmitted into NK cells disturbed PRR recognition, especially RIG-I, inducing NK cell dysfunction in CHB patients [22].

Stimulator of IFN genes (STING) is an endoplasmic-reticulum (ER)-membrane downstream adaptor protein of an intracellular DNA sensor, which can bind and activate TBK1 kinase and then induce the phosphorylation of several transcription factors, such as IRF3, and type I IFNs production, playing a crucial role in anti-virus and anti-tumor responses [24,25,26]. Current research has indicated that cGAS-STING signal activation could increase the levels of NKG2D ligands on tumor cells, which in turn activated the NK cell response [27,28]. However, the deficiency of STING impaired the spontaneous rejection to NK cell-sensitive tumor cells, such as B16-BL6 melanoma cells, which was independent of the NKG2D signal [29]. It has been suggested that HBV polymerase could inhibit the innate DNA-sensing signal by disrupting K63-linked ubiquitination of STING [30], and lacking STING expression in hepatocytes further revealed the incompetent type I IFN response to HBV [31]. Therefore, STING signal is crucial for both NK cell activation and anti-HBV response. However, the expression and regulation of STING in NK cells exposed to CHB infection have not been illustrated.

The purpose of this study was to clarify the mechanism of STING signal dysfunction in NK cells during CHB infection, and the relationships among STING, STAT3 and HBsAg were also discussed based on detection of STING levels of NK cells in CHB patients.

## 2. Result

### 2.1. STING Expression Was Suppressed in NK Cells from HBeAg-Negative CHB Patients

NK cells display a crucial immunoregulatory function during chronic HBV infection, but many studies have indicated that NK cell functions were impaired in CHB patients [22,32,33,34,35]. Martin K Thomsen, etc. illustrated that a lack of STING in hepatocytes dampened the anti-virus response, which facilitated persistent HBV infection [31]. To examine whether HBV infection influences the STING signal in NK cells, we firstly detected the STING levels in CD3^-^CD56^+^, CD3^-^CD56^bright^ and CD3^-^CD56^dim^ NK cells from healthy donors and HBeAg-negative CHB patients. We found that STING was significantly lower in NK cells from HBeAg-negative CHB patients than in NK cells from healthy donors (Figure 1A), but no significant differences between CD3^-^CD56^bright^ and CD3^-^CD56^dim^ were observed. In addition, we observed that STING expression in NK-92 cells could be downregulated by incubating them with CHB patient serum (Figure 1B) and an HBV-positive HepG2.1.15 cell supernatant (Figure 1C). These findings suggested that HBV components might impair NK cell function by inhibiting DNA sensor STING in HBeAg-negative CHB patients.

### 2.2. Dampened Expression of STING Correlated with NK Cell Dysfunction in HBeAg-Negative CHB Patients

Subsequently, we analyzed the influence of the STING signal pathway on the activation of NK cells from HBeAg-negative CHB patients. As shown in Figure 2A, the CD107a and Granzyme B levels in NK cells from healthy donors could be significantly upregulated by 2′3′-cGAMP treatment, while the NK cells from HBeAg-negative CHB patients showed low responsiveness to 2′3′-cGAMP stimulation. To better uncover the effect of HBV components on STING activation in NK cells, we further incubated NK-92 cells with the serum of HBeAg-negative CHB patients. Consistently, the results indicated that HBV-associated components could suppress NK cell responsiveness to STING agonist treatment (Figure 2B). These results indicated that the decreased expression of STING disturbs DNA sensing in NK cells from HBeAg-negative CHB patients.

### 2.3. HBsAg Downregulated STING Expression in NK Cells

HBsAg is one of the major components and diagnostic indicators of HBV. Previously, we found that HBsAg could impair NK cell function directly [36]. In order to elucidate the relationship between HBsAg and STING, we analyzed the STING levels of NK cells and serum HBsAg levels in HBeAg-negative CHB patients. As shown Figure 3A, the STING expression level of NK cells (CD3^-^CD56^+^, CD3^-^CD56^bright^ and CD3^-^CD56^dim^ cells) was negatively associated with HBsAg level. In addition, the STING expression of NK-92 cells was significantly downregulated by HBsAg treatment (Figure 3B). Meanwhile, the upregulation of CD107a and Granzyme B induced by 2′3′-cGAMP was eliminated by HBsAg incubation (Figure 3C). Thus, HBsAg is an important component of HBV, repressing STING expression and activation of NK cells.

### 2.4. STAT3 Regulated STING Expression in NK Cells

Signal transducers and activators of transcription 3 (STAT3) is a positive regulator of NK cells [35]. Our previous studies demonstrated that HBsAg suppressed NK cell function via STAT3 signaling [35]; thus, we tried to clarify the relationship between STING and STAT3. The correlation analysis showed that STING level was positively associated with STAT3 level in NK cells (CD3^-^CD56^+^, CD3^-^CD56^bright^ and CD3^-^CD56^dim^ cells) of HBeAg-negative CHB patients (Figure 4A). Furthermore, we observed that STING expression was upregulated in STAT3-over expression NK-92 cells and downregulated in STAT3-knockdown NK-92 cells (Figure 4B). Meanwhile, the expression of CD107a and Granzyme B was increased in NK-92 cells treated with 2′3′-cGAMP, while STAT3-knockdown diminished the responsiveness of NK-92 cells to 2′3′-cGAMP stimulation (Figure 4C). These findings illustrate that STAT3 is an upstream molecule of STING that positively regulates STING expression in NK cells.

### 2.5. STAT3 Directly Regulated STING Transcription in NK Cells

STAT3 is a vital transcription factor that directly binds to the promoters and regulates the mRNA levels of target genes. As shown in Figure 5A, STING mRNA levels were downregulated in NK cells of HBeAg-negative CHB patients. In addition, STAT3-knockdown could downregulate STING mRNA levels in NK-92 cells, while STING mRNA levels in STAT3-overexpressing NK-92 cells were significantly upregulated (Figure 5B). Based on these observations, we further determined whether STAT3 directly binds to the promoter of STING. With an upstream sequence of STING genes, we predictively analyzed 18 binding sites using the JASPAR database, and then we designed five pairs-specific primers (Appendix A) for the sequence (Appendix A) identification of anti-STAT3 Ab pull-down chromatin of NK-92 cells. As shown in Figure 5C, two DNA fragments were detected in p-STAT3 immunoprecipitated chromatin, with sequence and position shown in Figure 5D. Thus, these data demonstrated that STAT3 positively regulates STING gene transcription by directly binding to the promoters.

### 2.6. STING Associated IFNα Response Was Inhibited in NK Cells of HBeAg-Negative CHB Patients

It has been reported that a lack of a STING-dependent type 1 IFN response to HBV DNAs hampers innate defense against HBV in hepatocytes [31]. To determine whether STING-mediated type I IFNs’ response was influenced by chronic HBV infection, we isolated primary NK cells from healthy donors and HBeAg-negative CHB patients, and then we stimulated these with 2′3′-cGAMP. The results showed that the level of IFN-α was increased in NK cells from healthy donors by 2′3′-cGAMP treatment, accompanied with the activation of IRF3 (Figure 6A,B). However, the increase of IFN-α was dampened in NK cells from HBeAg-negative CHB patients (Figure 6A), and it was accompanied by the inhibition of phosphorylated IRF3 (Figure 6B). Consistent with this, STAT3 knockdown significantly decreased IFN-α response in NK-92 cells, accompanied by the suppressed activation of IRF3 (Figure 6C,D). Thus, STING-associated IFN-α expression was inhibited in NK cells from HBeAg-negative CHB patients.

## 3. Discussion

HBV is a kind of small DNA virus, which can evade immune surveillance and induce chronic infection in patients, but the mechanism has not yet been fully described. STING is a shared adaptor of both RIG-I and cGAS, playing a crucial role in exogenous RNA and DNA recognition of the innate immune system [37]. Studies have demonstrated that STING deficiency in hepatocytes significantly impairs the anti-virus response, which partly explains hepatocyte immune tolerance to HBV infection [31,38]. Our previous study indicated that exosomes could mediate HBV transmission into NK cells and induce NK cell dysfunction, accompanied by the inhibition of RIG-I recognition and TLRs [22]. However, the mechanism through which the DNA sensing pathway was not initiated by HBV DNAs is still unclear. In this study, we firstly found that STING was expressed in peripheral blood NK cells, the levels of which were significantly decreased in HBeAg-negative CHB patients (Figure 1A). Therefore, we speculated that STING may play crucial roles in HBV tolerance and DNA sensing dysfunction in NK cells.

The innate immune STING signal defends against viruses and cancers potently and rapidly, accompanied by the phosphorylation of the transcription factor IRF3 by kinase TBK1 and then the production of type I IFNs [39,40,41]. Mengze Lv etc. indicated that a STING agonist promoted DC and macrophage maturation and antigen presentation to activate CD8^+^T cells [42]. A STING agonist also enhanced NK cell activation in the clearance of tumor cells [43]. Consistent with this, the degranulation molecules of peripheral NK cells from healthy donors were significantly activated by a STING agonist, but no significant response was shown in NK cells from HBeAg-negative CHB patients. These results verified our hypothesis that low STING levels result in impaired DNA sensing and activation of NK cells in HBeAg-negative CHB patients (Figure 2). Thus, STING activation is important for NK activation in HBeAg-negative CHB patients.

STING levels of NK cells were decreased both in NK cells from HBeAg-negative CHB patients and in NK-92 cells incubated with CHB serum, indicating that a common component mediated the reduction of the STING levels in NK cells. In our and other previous studies, no HBV receptors, such as NTCP and ASGPR, were detected on NK cells, but HBsAg could inhibit NK cell function by direct cell surface binding and induce the differentiation of IL-10^+^ regulatory NK cells [33,36]. Here, we observed that STING levels in peripheral NK cells were negatively associated with serum HBsAg levels in CHB patients (Figure 3). Significantly, this phenomenon was more remarkable in male CHB patients than in female patients, but the age (over 40 years or low) did not influence the correlation between STING and HBsAg levels (Appendix A). Thus, HBsAg is likely involved in STING-associated NK cell recognition and activation during chronic HBV infection, suggesting that NK cell dysfunction might increase the risks of HBV recurrence and hepatocarcinogenesis in HBeAg-negative CHB patients, especially male patients.

STAT3 is a multiple function regulator of cells by signal transduction and activation transcription of its downstream genes. STAT3 signaling of tumor cells interacted with the STING signaling-associated immune response, and these two signals played opposing effects in the tumor microenvironment [44]. STAT3 activation attenuated STING-induced anti-tumor immunity, while the STING agonist induced an immune response that in turn restricted STAT3 activation in tumor cells [45,46]. However, the crosstalk between these two molecules has not been explored in the same cell, especially in NK cells. Previously, we found that STAT3 not only positively regulated the proliferation of NK cells, but also enhanced the expression of activating receptors, such as NKG2D and NKp46, as well as the function molecules, such as CD107a, Granzyme B, perforin and IFN-γ [35]. However, STAT3 levels in NK cells were decreased by HBsAg, which resulted in NK cell dysfunction in CHB patients. Here, STAT3 was positively associated with STING levels in NK cells of CHB patients, which was not influenced by gender and age (Appendix A), and STAT3 knockdown decreased the STING levels and inhibited NK cell activation (Figure 4). Interestingly, we confirmed that p-STAT3 directly bound to the promoter of STING and regulated the mRNA levels of STING in NK cells (Figure 5). Thus, different from the relationship between STAT3 and STING in the tumor microenvironment, STAT3 acts as a positive regulator in STING-associated activation of NK cells during CHB infection.

STING is one of the most effective alternative targets for HBV therapy. On the one hand, STING activated the type I IFNs’ response directly, which could enhance the other immune responses to defend against the virus. Martin et al. reported that overexpression of STING in hepatocytes recovered the innate response to HBV [31]. Estefania et al. indicated that AdrA, an inducer of STING-associated IFNs’ response, presented a highly efficient anti-HBV effect in HBV transgenic mice and adenovirus-associated virus (AAV)-HBV carrier mice [47]. On the other hand, as an adjuvant, STING ligand could enhance humoral and cellular immune responses to the HBV-vaccine and overcome immune tolerance in HBs-tg mice [48,49]. Although STING-associated IFNα responses were dampened in NK cells during CHB infection (Figure 6), STING expression may be recovered with STAT3 activation, such as IL-21 therapy. Thus, STAT3 activation may recover the STING response in NK cells of CHB patients, and STING or STAT3 might be an important candidate immune-activator for the treatment of CHB.

In conclusion, STING positively regulated the DNA sensing response in NK cells. The presence of HBsAg inhibited the STING expression and signal by inactivation of STAT3, and STAT3 acted as a positive transcription factor that directly bound to the promoter of STING. HBsAg-mediated dysfunction of NK cells might increase the risks of HBV recurrence and hepatocarcinogenesis in the inactive HBV carriers (HBsAg-positive and HBeAg-negative HBV carriers) who were not considered to need treatment in clinical settings. Therefore, our findings revealed a new mechanism of NK cell dysfunction in DNA sensing during CHB infection and provided a candidate target for HBV therapy.

## 4. Materials and Methods

### 4.1. Patient Samples

Peripheral blood samples and clinical data were collected during the regular follow-up of CHB patients from Qilu hospital of Shandong university. Informed consent was obtained from all participants, according to the guidelines and regulations of the Ethics Committee of Shandong University. All patients had no autoimmune disease and were negative for other viral infections, and the healthy donors had no history of liver disease. The clinical characteristics of the patients are shown in Table 1.

### 4.2. Cell Culture and Transfection

The 293T cells, HepG2.2.15 cells (HBV positive) and HepG2 cells were cultured in DMEM medium (GIBCO/BRL, Grand Island, NY, USA) with 10% FBS. The supernatant of HepG2.2.15 cells and HepG2 cells was collected every two days with culture medium replacement. NK-92 cells (human NK cell line) were cultured as described previously [35]. PBMCs were isolated from the peripheral blood of HBeAg-negative CHB patients and healthy donors by Ficoll-plus (P4350, Solarbio Life Science, Beijing, China). Primary human NK cells were purified by a negative selection MojoSortTM Human NK Cell Isolation Kit (Cat#480053, Biolegend, San Diego, CA, USA) and maintained in RPMI-1640 with 10% FBS and 100 U/mL recombinant human IL-2 (rhIL-2, Changchun Institute of Biological Products, China). STAT3-knockdown and STAT3-overexpressed NK-92 cell lines were respectively established, as previous described [35].

### 4.3. RNA Isolation and Quantitative Real-Time PCR (qRT-PCR)

RNA was isolated with TRzol™ Reagent (Cat#15596026, InvitrogenTM, Carlsbad, CA, USA), according to procedure of the manufacturer. Total RNA was normalized to 1 µg for cDNA generation using M-MLV Reverse Transcriptase (Cat#28025013, InvitrogenTM, Carlsbad, CA, USA). Quantitative RT-PCR was performed by Faststart Universal SYBR Green Master (Cat# 4913914001, Roche, Mannheim, Germany). The primer sequences are shown in Appendix A.

### 4.4. Western Blotting

NK-92 cells and primary NK cells were stimulated with 2′3′-cGAMP (10 μg/mL) (tlrl-nacga23, InvivoGen, La Jolla, CA, USA) for 12 h and collected by centrifugation. Total protein was extracted by an RIPA lysis buffer (Cat#P0013C, Beyotime Biotechonology, Shanghai, China) with 1 mM phenlymethylsulfonyl (PMSF). The total protein (30 µg/lane) was separated by SDS-PAGE and transferred to the PVDF membranes (Millipore, Billerica, MA, USA). The membrane was blocked in 5% Skim milk powder TBS solution and incubated with Anti-IRF3 (D83B9) rabbit mAb (CS4302#, Cell Signaling Technology, Danvers, MA, USA), anti-phospho-IRF3 (E7J8G) rabbit mAb (CS37829#, Cell Signaling Technology, Danvers, MA, USA) and anti-GAPDH rabbit mAb (R20006, Abmart, Shanghai, China) at 4 °C overnight. HRP-goat-anti-rabbit (A0208, Beyotime Biotechonology, Shanghai, China) and HRP-goat-anti-mouse (A0216, Beyotime Biotechonology, Shanghai, China) antibodies were used as secondary antibodies. The specific protein bands were visualized by an enhanced chemiluminescence system (Millipore, Billerica, MA, USA).

### 4.5. Surface and Intracellular Immunostaining for FACS

To detect CD107a molecules, NK cells were stimulated with 2′3′-cGAMP(10 μg/mL) (tlrl-nacga23, InvivoGen, Toulouse, France) for 6 h. In the last 4 h, NK cells were incubated with anti-CD107a antibody for 1 h, and then 10 µg/mL Brefeldin A (Cat#420601, Biolegend, San Diego, CA, USA) and 6 µg/mL monensin (475895, Sigma–Aldrich, St. Louis, MO, USA) were added. NK cells were collected 3 h later, and then they were stained with anti-CD3 and anti-CD56 antibodies for 1 h at 4 °C. To detect the intracellular molecules, the NK cells were fixed and permeabilized, and then they were incubated with the antibodies listed in Appendix A. For STING detection, NK-92 cells and primary NK cells were collected and incubated with the primary anti-TMEM173 antibody (ab189430, Abcam, Burlingame, CA, USA) for 1 h at 4 °C after permeabilizing and blocking, then they were stained with Alexa Fluor 700-conjugated goat anti-Rabbit IgG (H + L) cross-adsorbed secondary antibody (A-21038, InvitrogenTM, Carlsbad, CA, USA) for 30 min at room temperature. All cells were analyzed by BD FACSAria or FACSCalibur (BD Biosciences).

### 4.6. Chromatin Immunoprecipitation

NK-92 cells were stimulated with rhIL-2 for 24 h after starvation, and then these cells were collected and treated according to the protocol of the Magna ChIP™ A/G Chromatin Immunoprecipitation Kit (17-10085, Millipore, Billerica, MA, USA). Phospho-Stat3 (Tyr705) (D3A7) XP^®^ rabbit mAb (CS #9145, Cell Signaling Technology, Danvers, MA, USA) was used for immunoprecipitation, and EasyTaq^®^ DNA Polymerase (AP111, Transgen Biotech, Beijing, China) was used for PCR detection.

### 4.7. Statistical Analysis

The data presented here were analyzed by GraphPad software (Inc. La Jolla, CA, USA). The statistically significant differences between two groups were determined by a *t*-test and one-way ANOVA or by a two-way ANOVA for multigroup analysis. The correlation between variables was analyzed by the Pearson coefficient.

## Figures and Tables

**Figure 1 ijms-22-07643-f001:**
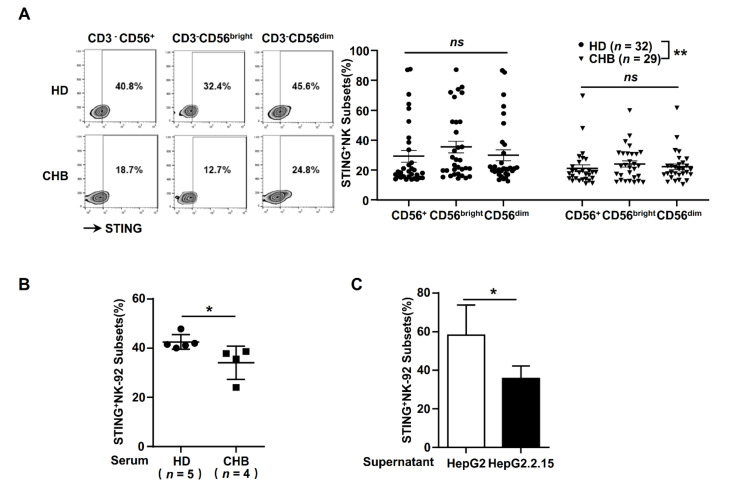
The STING level of NK cells was significantly decreased in CHB patients. (**A**) The ratio of peripheral STING^+^ NK cells (CD3^-^CD56^+^, CD3^-^CD56^bright^ and CD3^-^CD56^dim^ cells) in CHB patients and HDs. The statistically significant difference between HD and CHB was determined by a two-way ANOVA. (**B**) STING expression of NK-92 cells were analyzed by flow cytometry, after incubation with serum of HD and CHB patients for 48 h. The serum was diluted with 20 times culture medium of NK-92 cells. (**C**) STING expression of NK-92 cells were analyzed by flow cytometry after incubation with supernatant of HepG2 and HepG2.2.15 for 48 h. Data are shown as the mean ± SEM. The statistically significant difference between two groups was determined by a *t*-test. ns, *p* > 0.05; * *p* < 0.05; ** *p* < 0.01. CHB, CHB patients; HD, healthy donors; NK, natural killer.

**Figure 2 ijms-22-07643-f002:**
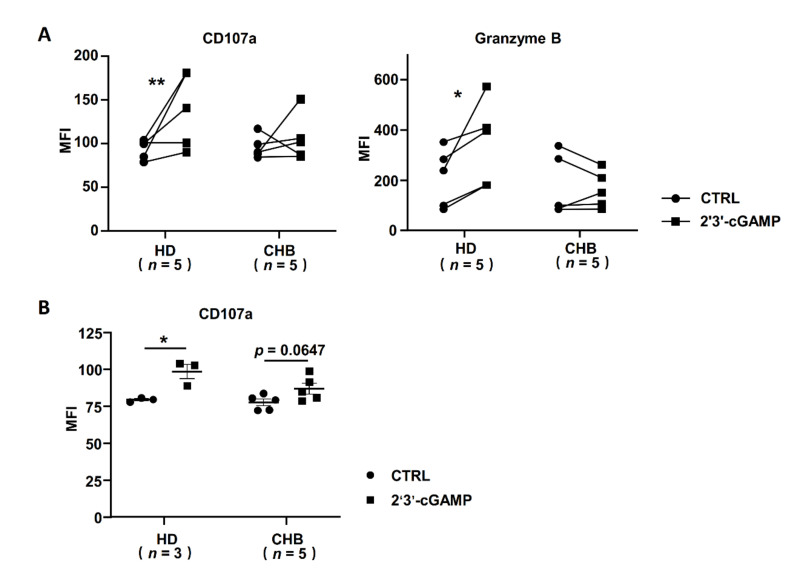
The response of STING in NK cells was significantly decreased in CHB patients. (**A**) The levels of CD107a and Granzyme B in NK cells of HD and CHB patients incubated with 2′3′-cGAMP for 6 h. (**B**) The CD107a levels of NK-92 cells incubated with serum of HD and CHB patients for 48 h and then stimulated with 2′3′-cGAMP for 6 h were analyzed by FACS. Data are shown as mean ± SEM. The statistically significant difference between two groups was determined by a *t*-test. * *p* < 0.05; ** *p* < 0.01. CHB, CHB patients; HD, healthy donors; NK, natural killer.

**Figure 3 ijms-22-07643-f003:**
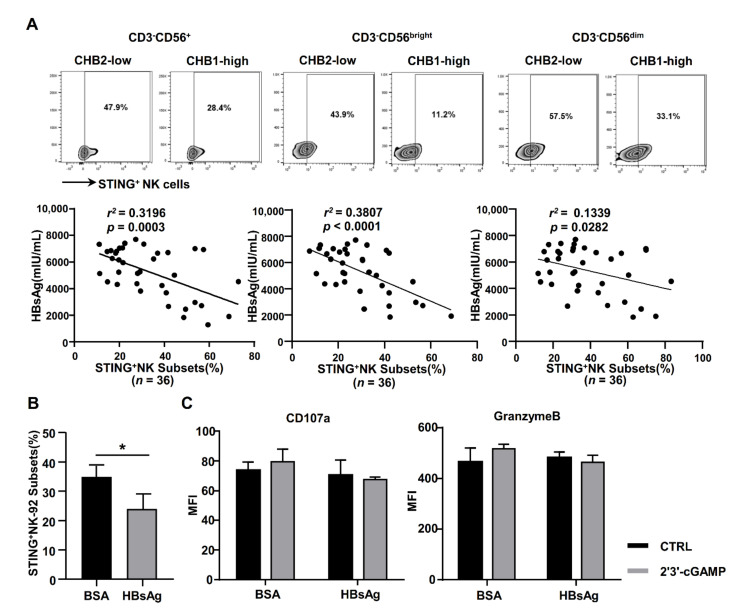
HBsAg inhibited the expression and response of STING in NK cells. (**A**) The correlation between serum HBsAg levels and STING expression levels in NK cells (CD3^-^CD56^+^, CD3^-^CD56^bright^ and CD3^-^CD56^dim^ cells) of CHB patients. (**B**) STING expression of NK-92 cells was analyzed by flow cytometry after incubation with HBsAg (20 μg/mL) for 48 h. (**C**) The CD107a and Granzyme B levels of NK-92 cells incubated with HBsAg were analyzed by FACS after stimulation with 2′3′-cGAMP (10 μg/mL) for 6 h. Data are shown as mean ± SEM from three independent experiments. The statistically significant difference between two groups was determined by *t*-test. The correlation between variables was analyzed by the Pearson coefficient. * *p* < 0.05. 0.04 < *r^2^* < 0.16, weak correlationship; 0.16 < *r^2^* < 0.36, moderate correlationship; 0.36 < *r^2^* < 0.64, strong correlationship; CHB, CHB patients; NK, natural killer.

**Figure 4 ijms-22-07643-f004:**
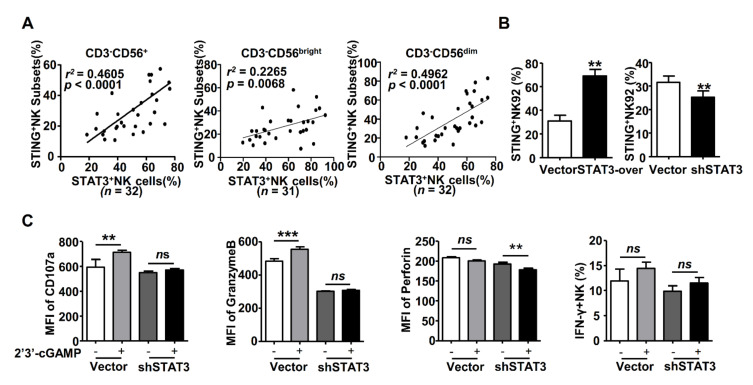
STAT3 regulated the expression and response of STING in NK cells. (**A**) The correlation between STING and STAT3 expression in the NK cells (CD3^-^CD56^+^, CD3^-^CD56^bright^ and CD3^-^CD56^dim^ cells) of CHB patients. (**B**) The levels of STING in STAT3-overexpressing and STAT3-knockdown NK-92 cells were analyzed by FACS. (**C**) CD107a and cytolysis markers’ expression in STAT3-knockdown or control NK-92 cells incubated with 2′3′-cGAMP (10 μg/mL) for 6 h. Data are shown as mean ± SEM from three independent experiments. The statistically significant difference between two groups was determined by a *t*-test and one-way ANOVA for multigroup analysis. The correlation between variables was analyzed by the Pearson coefficient. ns, *p* > 0.05; ** *p* < 0.01; *** *p* < 0.001. 0.04 < *r^2^* < 0.16, weak correlationship; 0.36 < *r^2^* < 0.64, strong correlationship. Vector, LMP or pCDH; shSTAT3, STAT3-knockdown NK-92 cells; STAT3-over, STAT3-overexpression NK-92 cells; NK, natural killer.

**Figure 5 ijms-22-07643-f005:**
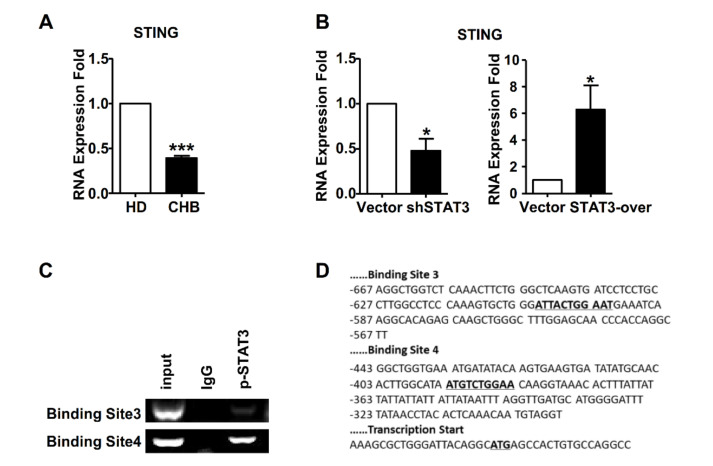
STAT3 regulated transcription of STING. (**A**) STING expression in NK cells isolated from HDs and CHB patients quantified by qPCR. (**B**) STING expression in STAT3-knockdown and -overexpressing NK-92 cells quantified by qPCR. (**C**,**D**) The predicted binding sites of STAT3 were verified by PCR on p-STAT3 antibody pull-down DNA of NK-92 cells, on the basis of JASPAR database analysis. Data are shown as mean ± SEM from three independent experiments. The statistically significant difference between two groups was determined by a *t*-test. * *p* < 0.05; *** *p* < 0.001. CHB, CHB patients; HD, healthy donors; Vector, LMP or pCDH; shSTAT3, STAT3-knockdown NK-92 cells; STAT3-over, STAT3-overexpression NK-92 cells; NK, natural killer.

**Figure 6 ijms-22-07643-f006:**
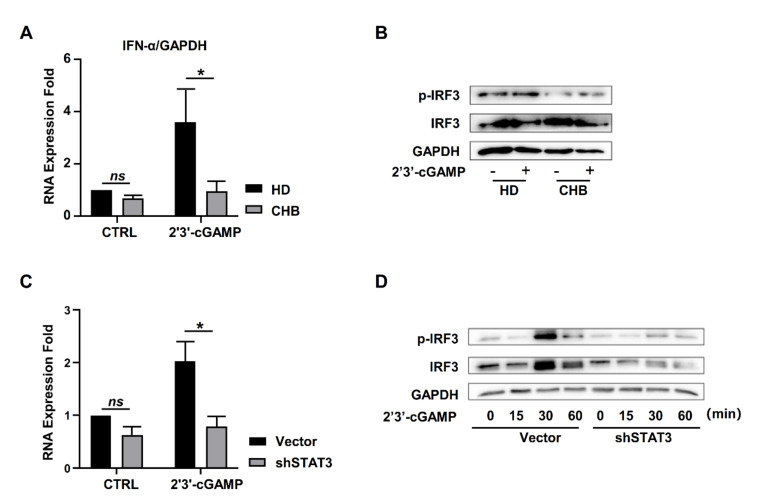
STING-associated IFNα response was inhibited in NK cells of HBeAg-negative CHB patients. (**A**) IFNα expression in NK cells isolated from HDs and CHB patients was quantified by qPCR after stimulation with 2′3′-cGAMP (10 μg/mL) for 12 h. (**B**) IRF3 and phosphorylated IRF3 in NK cells isolated from HDs and CHB patients were quantified by western blotting after stimulation with 2′3′-cGAMP (10 μg/mL) for 30 min. (**C**) IFNα expression in STAT3-knockdown NK-92 cells was quantified by qPCR after stimulation with 2′3′-cGAMP (10 μg/mL) for 12 h. (**D**) IRF3 and phosphorylated IRF3 in STAT3-knockdown NK-92 cells were quantified by western blotting after stimulation with 2′3′-cGAMP (10 μg/mL) for 15, 30 and 60 min. Data are shown as mean ± SEM from three independent experiments. The statistically significant difference between two groups was determined by a *t*-test. ns, *p* > 0.05; * *p* < 0.05. CHB, CHB patients; HD, healthy donors; Vector, LMP or pCDH; shSTAT3, STAT3-knockdown NK-92 cells; STAT3-over, STAT3-overexpression NK-92 cells; NK, natural killer.

**Table 1 ijms-22-07643-t001:** Clinical data of healthy donors and CHB patients in the study.

	HD	CHB
Number	90	107
Gender (M/F)	42/48	61/46
Age (years)	46.84 ± 11.51	41.14 ± 11.05
HBsAg (Pos/Neg)	0/90	107/107
HBeAg (Pos/Neg)	0/90	0/107

Abbreviations: CHB, chronic hepatitis B; F, female; HD, healthy donor; M, male; Neg, negative; Pos, positive.

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
