# Peer review of "HBsAg Dampened STING Associated Activation of NK Cells in HBeAg-Negative CHB Patients"

_ijms, 2021, doi:10.3390/ijms22147643_

Round 1

Reviewer 1 Report

This manuscript revealed HBsAg directly suppressed STAT3-STING signaling axis and impaired NK cytotoxicity and type I IFN production. Their results are important for understanding NK dysfunction in chronic hepatitis B virus infection. Although all experiments are well-designed, there are some points to be corrected.

Comments

Major points

  1. Authors classified NK cells as CD3-CD56+ cells, but, as you know, NK cells can be divided by surface markers, such as CD16 and CD57. Because these two subsets are known to have different immune activity, authors should analyze each NK subset. I believe this increase the value of their finding. If you did not stain these subset markers in your experiments, the expression level of CD56 can be used for roughly dividing NK population, please try.
  2. For the convenience of journal readers, I recommend authors add the number of experiments and subjects for each figure and the description of error bar of bar graphs in all figures.
  3. I afraid the number of subjects is small for statistical analysis in figure 1B, 2A and 2B. In table 1, authors analyzed over 90 subjects for each group, I wonder why authors examined so few subjects of groups in their analysis. If possible, please examine all subjects for more correctly evaluating your results.
  4. In correlation analysis, when dividing subjects by their age (over 40 or low) or gender, will correlation coefficiencies be changed?

Minor point

In reference 19, publication year is repeated.

Reviewer 2 Report

The authors stated clearly what study found and how they did it. Appropriate and key studies are included. The research question also justified given what is already known about the topic.

The variables are well defined and measured appropriately. The study methods are valid and reliable. There are enough details provided in order to replicate the study.

The data is presented in an appropriate way. The text in the results add to the data and it is not repetitive. Statistically significant results are clear. Results are discussed from different angles and placed into context without being over-interpreted.

The conclusions answer the aim of the study. The conclusions are supported by references and own results.

Specific comments on weaknesses of the article and what could be improved:

Major points  - none

Minor points

  1. Please, state the limitations of the study
  2. Could you please discuss the clinical implications of the results
  3. Remove the text "figure 1" in Fig. 1 graphics. This is valid for the rest of the figures.
  4. Figure 4 - in the caption - What does it mean "The statistic significant"?
  5. Which results are with practical meaning?

Round 2

Reviewer 1 Report

I agree with acceptance of this manuscript. Congratulation!